# Multipath-Assisted Ultra-Wideband Vehicle Localization in Underground Parking Environment Using Ray-Tracing

**DOI:** 10.3390/s25072082

**Published:** 2025-03-26

**Authors:** Shuo Hu, Lixin Guo, Zhongyu Liu, Shuaishuai Gao

**Affiliations:** School of Physics, Xidian University, Xi’an 710071, China; shu@stu.xidian.edu.cn (S.H.); ssgao@stu.xidian.edu.cn (S.G.)

**Keywords:** vehicle positioning, time-of-arrival, angle-of-arrival, time-difference-of-arrival, non-line-of-sight, ray-tracing

## Abstract

In complex underground parking scenarios, non-line-of-sight (NLOS) obstructions significantly impede positioning signals, presenting substantial challenges for accurate vehicle localization. While traditional positioning approaches primarily focus on mitigating NLOS effects to enhance accuracy, this research adopts an alternative perspective by leveraging NLOS propagation as valuable information, enabling precise positioning in NLOS-dominated environments. We introduce an innovative NLOS positioning framework based on the generalized source (GS) technique, which employs ray-tracing (RT) to transform NLOS paths into equivalent line-of-sight (LOS) paths. A novel GS filtering and weighting strategy to establish initial weights for the nonlinear equation system. To combat significant NLOS noise interference, a robust iterative reweighted least squares (W-IRLS) method synergizes initial weights with optimal position estimation. Integrating ultra-wideband (UWB) delay and angular measurements, four distinct localization modes based on W-IRLS are developed: angle-of-arrival (AOA), time-of-arrival (TOA), AOA/TOA hybrid, and AOA/time-difference-of-arrival (TDOA) hybrid. The comprehensive experimental and simulation results validate the exceptional effectiveness and robustness of the proposed NLOS positioning framework, demonstrating positioning accuracy up to 0.14 m in specific scenarios. This research not only advances the state of the art in NLOS positioning but also establishes a robust foundation for high-precision localization in challenging environments.

## 1. Introduction

Accurate and reliable vehicle positioning has become a fundamental requirement in modern intelligent transportation systems (ITSs) [1,2,3]. With the emergence of 5G/6G integrated sensing and communication systems (ISACs), precise positioning services have become key enablers for various applications, such as collision avoidance, cooperative driving, and intelligent navigation [4,5]. While significant progress has been made in vehicle positioning for open-road scenarios, underground parking environments pose unique challenges due to their confined spaces, severe signal attenuation [6,7], and limited access to satellite-based positioning systems like GPS [8,9,10].

In underground parking garages, accurate and continuous vehicle positioning is crucial for improving user experience, enhancing parking efficiency, and supporting safety-critical applications such as automated parking, vehicle retrieval systems, and efficient parking management [11,12,13]. As the development of autonomous driving and connected vehicles continues to accelerate, precise vehicle positioning in such environments not only improves the reliability of location-based services (LBSs) but also minimizes potential risks caused by positioning errors, which could otherwise lead to inefficient parking operations or even accidents [14,15]. However, underground parking environments pose significant challenges for positioning technologies, including signal blockage, multipath effects, and non-line-of-sight (NLOS) propagation, which must be addressed to deliver precise and reliable location information [16]. Furthermore, the demand for seamless integration with intelligent navigation systems highlights the need for robust and scalable solutions tailored specifically to underground environments [17,18].

In recent years, researchers have proposed various solutions to address the problem of vehicle localization in non-line-of-sight (NLOS) scenarios. Vehicle localization technologies in obstructed environments are primarily based on platforms such as ultra-wideband (UWB), Wi-Fi, BLE, LiDAR, and visible light [19,20,21,22,23,24,25,26]. UWB is a short-range wireless communication technology that offers centimeter-level positioning accuracy and demonstrates excellent performance in complex multipath environments. Moreover, UWB technology employs low-power signal transmission, provides strong anti-interference capabilities, and is minimally impacted by electromagnetic interference from the environment, making it particularly suitable for NLOS localization applications [27,28,29].

NLOS localization techniques based on UWB generally focus on two main aspects: multipath mitigation techniques, represented by NLOS site identification [30,31,32,33] and NLOS error compensation [34,35,36], and multipath exploitation techniques based on geometric mapping [37,38,39,40,41,42]. Multipath mitigation techniques rely on empirical data to establish statistical characteristics of NLOS errors. Channel parameters such as RSS, root mean square (RMS) delay, and RMS angle are used to distinguish between line-of-sight (LOS) and NLOS sites [43,44]. In [45], the researchers constructed a fingerprint database by combining UWB and Wi-Fi and utilized the density peak clustering technique to differentiate LOS and NLOS sites, thereby improving localization accuracy. To better characterize the distribution of NLOS error features, some scholars have adopted machine learning techniques [46,47,48]. For instance, in [49], the researchers applied the long short-term memory (LSTM) method to train and identify raw channel impulse response (CIR) data collected from UWB channels, enhancing the accuracy of time-of-arrival (TOA) algorithms in NLOS scenarios. In [50], convolutional neural networks (CNNs) and LSTM techniques were simultaneously applied for NLOS site identification, achieving an identification accuracy of over 82%.

While deep learning methods have shown promising performance in NLOS environments, their reliance on large amounts of labeled data and high computational complexity remain significant challenges for practical applications. As a classical state estimation method, the Kalman filter (KF) has emerged as a vital tool for addressing NLOS localization problems due to its real-time performance and robustness in dynamic systems [51,52,53]. In [54], the researchers proposed a robust extended Kalman filter (REKF) algorithm combined with distance constraint, which improved localization accuracy in NLOS scenarios. The work in [55] introduced a novel localization strategy based on constrained square root unscented Kalman filtering (CSRUKF) and the robust Taylor series (RTS) algorithm, further enhancing the robustness of NLOS error mitigation.

Although the aforementioned traditional localization methods have demonstrated significant effectiveness in improving vehicle localization accuracy in NLOS scenarios, their heavy reliance on empirical data often leads to high costs in data acquisition and model training. To overcome this limitation, researchers have shifted their focus to exploiting multipath information in complex environments for localization. In [56], scholars utilized the single-reflection mechanism of scatterers to identify virtual sources (VSs) and incorporated a two-step weighted least squares (TSWLS) method to enhance localization accuracy in NLOS scenarios. The work in [57] expanded the concept of VSs to include multiple scattering propagations, combining angle-of-arrival (AOA) and TOA hybrid localization methods to improve the applicability and robustness of VS techniques in NLOS scenarios. To further enhance algorithm accuracy and efficiency, in [58], the researchers applied VS techniques to hybrid AOA and time-difference-of-arrival (TDOA) localization and proposed an initial position estimation method. This approach, combined with the TSWLS technique, achieved effective localization under NLOS conditions.

The VS technique has demonstrated remarkable localization performance in NLOS scenarios; however, it still faces several significant limitations. Many existing methods do not fully exploit the complete information provided by channel propagation and lack a thorough analysis from the perspective of electromagnetic wave behavior. While TOA-based cooperative localization algorithms can precisely determine the positions of virtual sources for each multipath, non-cooperative methods such as AOA or TDOA struggle to achieve comparable accuracy. This shortcoming greatly diminishes their performance in complex multipath environments, making it challenging to meet the requirements of high-precision localization. In addition, most approaches rely on a single localization algorithm, which proves inadequate for scenarios like underground parking garages with complex reflections and diffractions. These environments, characterized by prominent multipath effects and highly intricate propagation paths [59], pose substantial challenges to the VS technique. Furthermore, the influence of modeling accuracy on algorithm performance is often neglected, further limiting the effectiveness of these methods in complex scenarios.

To address the aforementioned issues, this paper proposes an NLOS vehicle localization-based service framework that utilizes state-of-the-art ray-tracing (RT) [60] and the generalized source (GS) technique. In the proposed method, an innovative GS filtering method based on geometric restriction conditions (GRCs) and a GS weighting method utilizing physical channel characteristics are introduced to improve the algorithm’s accuracy and applicability in NLOS propagation environments. Additionally, a robust IRLS method with initial weights (W-IRLS) is proposed to solve equations containing NLOS noise. Meanwhile, the RT-VLBS platform, based on a UWB system, supports TOA and AOA/TOA cooperative localization modes, as well as AOA and AOA/TDOA non-cooperative localization modes. The contributions of this paper can be summarized as follows:We propose an innovative ray-tracing vehicle localization-based service (RT-VLBS) framework that leverages multipath assistance through the integration of the GS technique and RT methodology. The framework effectively converts NLOS paths into valuable positioning information, achieving robust and high-precision localization in NLOS environments.A novel GS filtering and weighting strategy is proposed to heuristically optimize the weights of NLOS nonlinear localization equations, substantially improving both the accuracy and reliability of the positioning algorithm.Extensive experiments using the UWB system in an underground parking garage, strategically designed to capture NLOS multipath propagation characteristics, comprehensively validated the effectiveness and reliability of RT-VLBS in challenging NLOS scenarios.To verify the RT-VLBS’s robustness and reliability, different measurement parameter errors and environmental geometric modeling errors in NLOS scenarios were simulated and analyzed.

The rest of this paper is organized as follows: Section 2 outlines the fundamental principles of GS under the assistance of RT, including GS generation, filtering, and weight calculation. Section 3 introduces the localization methods utilized by the RT-VLBS platform, encompassing four localization modes: AOA, TOA, joint AOA/TOA, and joint AOA/TDOA, along with the application of the W-IRLS method in each mode. Section 4 details the experimental validation conducted in an underground parking garage under NLOS conditions, using a platform equipped with UWB technology to demonstrate the effectiveness of the RT-VLBS. Section 5 presents extensive simulation experiments to assess the robustness of the RT-VLBS platform under varying measurement errors and environmental geometric modeling uncertainties. Finally, the findings of this study are discussed, along with its limitations.

## 2. Basic Principles of RT-Assisted Generalized Sources

Figure 1 illustrates the main workflow of the RT-VLBS localization method, which consists of three key algorithms: GS generation, GS filtering, and GS weighted localization. In the GS generation algorithm, an advanced RT method is employed for channel modeling based on the measurement data from localization stations and an accurate environmental model. This method converts NLOS paths into line-of-sight (LOS) paths, resulting in an initial set of GSs. Due to the lack of effective prior information, the initial GS set contains a large number of invalid values. To address this, a GS filtering method based on geometric restriction conditions is introduced to effectively eliminate erroneous GSs. In the GS weighted localization algorithm, an initial weight matrix computation method based on residual weighting and a robust localization algorithm are proposed to enhance localization accuracy and robustness in NLOS environments. In the following sections, we provide a detailed description of these three core algorithms.

### 2.1. GS Generation

In the NLOS environment of an underground parking garage, electromagnetic waves between the mobile stations (MSs) and base stations (BSs) are typically blocked by obstacles, resulting in NLOS conditions in most scenarios. As illustrated in Figure 2, $BS_{1}$, $BS_{2}$, and $BS_{3}$ represent three base stations, while MS is a mobile station placed on a sedan. In this scenario, the MS is in an NLOS state with respect to the BSs. The positioning signal is transmitted by the MS and propagates through NLOS paths to reach the BSs. The RT algorithm is employed to transform NLOS paths into equivalent LOS paths. In this process, rays are emitted from the BSs to search for the GS locations. In the scenario depicted in Figure 2, with the RT search depth set to 1, the GS of $BS_{1}$, $BS_{2}$, and $BS_{3}$ are derived as $GS_{1}$, $GS_{2}$, and $GS_{3}$. Among them, $GS_{1}$ is a diffraction GS, while $GS_{2}$ and $GS_{3}$ are reflection GSs. By adjusting the search depth, GSs at different multipath depths can be obtained.

Figure 3a illustrates the propagation path of a signal transmitted from the mobile station (denoted as MS), which undergoes two reflections between obstacles before reaching the localization station (denoted as BS). The total propagation path length is t, and the received angle information is θ.

The GS generation algorithm employs an RT approach. In particular, a ray-tube-based shooting and bouncing ray (SBR) method, which falls under the RT framework, is employed in the GS generation algorithm. A vector-based electronic map is incorporated into the RT method to precisely compute ray–environment occlusion and assess visibility conditions. The construction modes of the GS can be categorized into angle-based modes (i.e., AOA, AOA/TOA, and AOA/TDOA localization modes) and non-angle-based modes (i.e., TOA localization mode). As depicted in Figure 3b, this method first generates ray tubes based on the AOA information received at the BS and subsequently performs path tracing. Moreover, during the construction process, TOA information is utilized to impose propagation distance constraints on the ray tubes, effectively filtering out $g_{1}$ and $g_{3}$. In the absence of AOA information, ray tubes are emitted isotropically in all directions, with each ray tube constrained by a propagation distance limit of t, as illustrated in Figure 3c.

Let the coordinates of the intersection point where a ray collides with an obstacle be denoted as $x_{n}={\left[x_{n},y_{n}\right]}^{T}$, and the direction of the central ray of the emerging ray tube be represented as ${\hat{x}}_{a}={\left[x_{a},y_{a}\right]}^{T}$. The minimum and maximum propagation distances of the ray tube are denoted as $t_{min}$ and $t_{max}$, respectively. In cases where the propagation path consists of line-of-sight (LOS) or reflection components, $t_{min}=0$. Conversely, for diffraction-based propagation paths, $t_{min}=t_{max}$. Hence, the location of the GS $x_{g}$ can be determined as follows:(1)$$x_{g}=x_{n}-{\hat{x}}_{a}\cdot \left(t_{max}-t_{min}\right)$$

According to Equation (1), if $x_{g}$ corresponds to a reflected GS, its position is the mirror image of the BS with respect to the environment. If $x_{g}$ corresponds to a diffracted GS, its position coincides with the diffraction point (i.e., the wedge). It is important to note that, although the RT algorithm can effectively trace reflected GSs and diffracted GSs within the geometry, the nature of diffraction renders the AOA information inapplicable to diffracted GSs. Moreover, considering the relatively low energy associated with diffraction, the search for diffracted GSs is conducted only once in the GS generation process.

### 2.2. GS Filtering

The proliferation of GSs is fundamentally determined by two key factors: environmental complexity and the depth of RT algorithm exploration. In complex scenarios such as underground parking garages, the multipath propagation between MS and BS frequently extends to first-order or higher, resulting in the generation of numerous GSs associated with each BS. While TOA-based localization modes can effectively filter out superfluous GSs through precise delay information, AOA-based and TDOA-based approaches face significant challenges due to the absence of reliable prior information for determining valid multipath propagation depths between MS and BS. This limitation leads to the persistence of numerous invalid GSs in the system. The resultant NLOS-induced noise can severely impact the localization performance, either by disrupting the convergence of localization equations or, in more severe instances, causing substantial deviations from the actual target position.

We propose an innovative filtering approach based on generalized source pairs (GSPs), formed by systematically pairing individual GSs. As illustrated in Figure 2, all initial GSs are systematically paired in sequence to form the corresponding GSPs. Given the initial number of $N_{0}$ GSs, this pairing process yields a total of N02. By leveraging the observation parameters of each GS, we construct an initial set of localization equations, comprising AOA, TOA, and TDOA equations. Notably, while AOA and TOA equations directly utilize parameters from the GSPs, TDOA equations require supplementary reference GSs beyond the paired structure.

Consider a GSP containing $GS_{i}$ and $GS_{j}$ with coordinates $a_{i}={\left[a_{i,x},a_{i,y}\right]}^{T}$ and $a_{j}={\left[a_{j,x},a_{j,y}\right]}^{T}$, respectively. The multipath information for $GS_{i}$ is characterized by $\left\{\varphi_{i},\tau_{i},d\tau_{i}\right\}$, while for $GS_{j}$, it is denoted as $\left\{\varphi_{j},\tau_{j},d\tau_{j}\right\}$, where φ represents the angle of arrival, τ indicates the propagation delay, and dτ denotes the delay difference. Given that the target position to be determined within the GSP is $x={\left[x,y\right]}^{T}$, localization equations can be constructed based on the multipath information of the GSP:(2)$$\left\{\begin{array}{c} \left(x-a_{i,x}\right)\tan\varphi_{i}=y-a_{i,y}\\ \left(x-a_{j,x}\right)\tan\varphi_{j}=y-a_{j,y} \end{array}\right.$$(3)$$\left\{\begin{array}{c} \left\|x-a_{i}\right\|=\sqrt{{\left(x-a_{i,x}\right)}^{2}+{\left(y-a_{i,y}\right)}^{2}}=c\tau_{i}\\ \left\|x-a_{j}\right\|=\sqrt{{\left(x-a_{j,x}\right)}^{2}+{\left(y-a_{j,y}\right)}^{2}}=c\tau_{j} \end{array}\right..$$(4)$$\left\{\begin{array}{c} \left\|x-a_{i}\right\|-\left\|x-a_{ref}\right\|=d\tau_{i}\\ \left\|x-a_{j}\right\|-\left\|x-a_{ref}\right\|=d\tau_{j} \end{array}\right.$$

Among these formulations, Equation (2) represents the AOA-based localization equations, Equation (3) describes the TOA-based localization equations, and Equation (4) characterizes the TDOA-based localization equations. In these equations,
$a_{ref}$
designates the coordinates of the reference GS within the GSP framework.

To optimize localization accuracy, we introduce a comprehensive framework that synergistically combines different localization parameters. The proposed system implements four distinct solution modes, each utilizing specific observation parameters: a standalone AOA-based mode, a standalone TOA-based mode, and two hybrid modes—AOA/TOA and AOA/TDOA. For each GSP, an initial solution is computed using the least squares (LS) methodology.

The algorithm implements two geometric restriction conditions (GRCs) to validate the GSP selections. The first GRC enforces that the GSP solution must be contained within a predefined feasible domain. The second GRC examines the geometric integrity of multipath reflection points, ensuring they do not intersect with environmental obstacles. For instance, as depicted in Figure 2, $BS_{3}$ generates a second-order reflection path that produces two GSs, $GS_{3}$ and $GS_{4}$, with corresponding reflection points $P_{1}$ and $P_{2}$. The geometric validity of any GSP containing $GS_{3}$ or $GS_{4}$ is determined by examining the line segments $P_{1}MS$ and $P_{2}MS$. The presence of any obstacles along these segments, as determined through ray-tracing-based occlusion detection, renders the corresponding GSP geometrically invalid.

Algorithm 1 delineates the systematic procedure for GS filtering. The process initiates with solving localization equations for each formed GSP. When a solution exists and satisfies the GRCs, the algorithm increments the weight counter of both constituent GSs within that GSP by 1. Following the comprehensive evaluation of all GSPs, the algorithm eliminates GSs with zero weight counts from further consideration. Since both angle and time delay measurements are subject to errors, the solutions within a GSP may fail to satisfy the GRCs under erroneous conditions, potentially leading to the erroneous elimination of valid GSs. To address this issue, the solutions obtained in this study are perturbed in four directions along the coordinate axis by a small displacement Δd (typically set to 0.1 m). A GSP is deemed invalid only if none of the five perturbed solutions satisfy the GRCs.
**Algorithm 1.** GS filtering algorithm**Precondition:** Generate all GS, with the total number denoted as $N_{0}$.Sequentially construct all possible ordered GS pairs, yielding N02 GSPs.**Foreach** GSP in GSPs        Formulate base Equations (2)–(4) and compute the initial solution x of the GSP through LS optimization.        If x exists and satisfies GRCs               Increment the weight count of the two GSs in the current GSP by 1.**        End If****End Foreach**Filter out GSs with zero weight count. The number of valid GSs denoted as $N_{gs}$.Proceed to subsequent processing steps.

### 2.3. GS Weighting

While the initial filtering process eliminates certain erroneous GSs, the environmental complexity introduces additional challenges, potentially retaining a substantial number of spurious GSs in the valid set. Under certain localization modes, such as the AOA localization mode, BSs only provide angular information. Due to the lack of reliable positioning data or the presence of significant measurement errors, the ray-tracing-based GS generation algorithm fails to determine the appropriate search depth, leading to the emergence of spurious GSs that cannot be filtered out by the GRCs. This ultimately degrades the performance of the positioning algorithm. Figure 2 illustrates this phenomenon through $GS_{3}$ and $GS_{4}$, which are mutually exclusive under physical propagation conditions: $GS_{3}$ is valid for a first-order reflection path between MS and $BS_{3}$, whereas $GS_{4}$ corresponds to a valid second-order reflection path. While only a single set of positioning data is measured by $BS_{3}$, only one of $GS_{3}$ and $GS_{4}$ can be valid. This ambiguity compounds with increasing search depth, leading to a proliferation of such invalid GSs.

To overcome this challenge, we propose a residual-based GS weighting methodology. As depicted in Figure 1, the approach initially employs hierarchical clustering to consolidate the preliminary GSP solutions into GSP clusters (GSPCs). Subsequently, RT techniques are employed to compute cluster central multipath information $M_{s}$ between the target position and $N_{BS}$ BSs, which can be formulated as follows:(5)$$M_{s}=\left\{\left(\varphi_{s,i},\tau_{s,i},d\tau_{s,i},p_{s,i},dp_{s,i}\right)|i\in \left\{1,2,\ldots ,N_{BS}\right\}\right\}$$
where φ denotes the angle of arrival, τ is the time of arrival, dτ represents the time difference in arrival, p  signifies the received signal strength (RSS), and dp indicates the RSS differential.

Similarly, the actual measured multipath information $M_{m}$ is defined analogously to $M_{s}$, with the subscript s replaced by  m.

Hence, the residuals for each GSPC can be obtained, which include the angular residual $R_{cl,j}^{\left(\varphi \right)}$, the time residual $R_{cluster}^{\left(\tau \right)}$, the time difference residual $R_{cluster}^{\left(d\tau \right)}$, the power residual $R_{cluster}^{\left(p\right)}$, and the power difference residual $R_{cluster}^{\left(dp\right)}$. These residuals are computed as follows:(6)$$R_{cl,j}^{\left(X\right)}=\sum_{i=1}^{N_{cl}}\left|X_{m_{i}}-X_{s_{i}}\right|,j\in \left\{1,2,\ldots ,N_{cl}\right\}$$
where X represents each of the aforementioned parameters, and $N_{cl}\;$ denotes the number of GSPCs.

The normalized residuals ${\hat{R}}_{cl,j}^{\left(X\right)}$ can be computed as follows:(7)$${\hat{R}}_{cl,j}^{\left(X\right)}=\frac{R_{cl,j}^{\left(X\right)}}{\underset{1\leq j\leq N_{cl}}{\max}R_{cl,j}^{\left(X\right)}}$$
where the denominator ensures all values are scaled relative to the maximum residual value. Hence, the normalized residuals under the four positioning modes can be computed as follows:(8)$${\hat{R}}_{cl,j}^{\left(AOA\right)}=w_{\varphi }{\hat{R}}_{cl,j}^{\left(\varphi \right)}+w_{p}{\hat{R}}_{cl,j}^{\left(dp\right)}$$(9)$${\hat{R}}_{cl,j}^{\left(TOA\right)}=w_{\tau }{\hat{R}}_{cl,j}^{\left(\tau \right)}+w_{p}{\hat{R}}_{cl,j}^{\left(p\right)}$$(10)$${\hat{R}}_{cl,j}^{\left(AOA/TOA\right)}=w_{\varphi }{\hat{R}}_{cl,j}^{\left(\varphi \right)}+w_{\tau }{\hat{R}}_{cl,j}^{\left(\tau \right)}+w_{p}{\hat{R}}_{cl,j}^{\left(p\right)}$$(11)$${\hat{R}}_{cl,j}^{\left(AOA/TDOA\right)}=w_{\varphi }{\hat{R}}_{cl,j}^{\left(\varphi \right)}+w_{\tau }{\hat{R}}_{cl,j}^{\left(\tau \right)}+w_{p}{\hat{R}}_{cl,j}^{\left(dp\right)}$$

Here, $w_{\varphi }$, $w_{\tau }$, and $w_{p}$ represent the weight coefficients for angular, delay, and power residual components, respectively. Weight sum constraints ($w_{\varphi }+w_{p}=1$ for AOA, $w_{\tau }+w_{p}=1$ for TOA, and $w_{\varphi }+w_{\tau }+w_{p}=1$ for hybrid cases) must be satisfied. Though typically assigned equal values, these weights can be dynamically adjusted based on measurement error variations.

The normalized weight expression for a GSPC is given as follows:(12)$${\hat{w}}_{cl,j}^{\left(Mode\right)}=\frac{1/{\hat{R}}_{cl,j}^{\left(Mode\right)}}{\underset{1\leq j\leq N_{cl}}{\max}\left(1/{\hat{R}}_{cl,j}^{\left(Mode\right)}\right)}$$
where Mode represents different localization modes, including AOA, TOA, AOA/TOA, and AOA/TDOA localization modes.

Following the computation of GSPC weights, the algorithm updates the weights of individual GSs within each cluster. The aggregation of all GS weights yields an initial weight matrix $W_{0}$, structured as a diagonal matrix $diag\left(w_{0,1},w_{0,2},\ldots ,w_{0,N_{gs}}\right)$, where $N_{gs}$ denotes the number of valid GSs.

## 3. Vehicle Localization Algorithm

### 3.1. Initial Solution Selection

Under NLOS conditions, the positioning equation solution exhibits high sensitivity to initial value selection, where an appropriate initial value is crucial for ensuring rapid convergence. The proposed algorithm designates the highest-weighted solution within the GSPC as the initial solution, expressed as $x^{\left(0\right)}={\left[x_{0},y_{0}\right]}^{\;T}$.

The positioning accuracy in AOA/TDOA hybrid scenarios is heavily dependent on the selection of the reference GS. A key challenge stems from the uncertainty of the specific multipath propagation between the reference BS and MS, which prevents the determination of the corresponding reference GS. This paper presents a heuristic approach to resolve this issue. The method employs RT algorithms to analyze multipath propagation between the initial solution of the highest-weighted GSPC and reference BS, subsequently selecting the GS associated with the minimum-delay multipath as the best reference GS.

### 3.2. Robust Localization Estimator

Various positioning equations can be formulated according to different observation parameters. Converting Equations (2)–(4) into their corresponding error forms yields the following:(13)$$\begin{array}{cc} \varphi_{i}=tan^{-1}\left(\frac{y-a_{iy}}{x-a_{ix}}\right)+e_{i}^{\left(\varphi \right)}, & i=1,2,\ldots ,N_{gs} \end{array}$$(14)$$\tau_{i}=\frac{\left\|x-a_{j}\right\|}{c}+e_{i}^{\left(\tau \right)},\;i=1,2,\ldots ,N_{gs}$$(15)$$\begin{array}{cc} d\tau_{i}=\frac{\left\|x-a_{i}\right\|-\left\|x-a_{1}\right\|}{c}+e_{i}^{\left(d\tau \right)}, & i=2,3,\ldots ,N_{gs} \end{array}$$

In the IRLS method, the linearization of the objective function is crucial for iterative position updating. At the k-th iteration, the initial position $x^{\left(k\right)}$ (obtained from the optimal cluster center in the first iteration $x^{\left(0\right)}$) serves as the linearization point. The error functions $e_{i}^{\left(\varphi \right)}$, $e_{i}^{\left(\tau \right)}$, and $e_{i}^{\left(d\tau \right)}$ are approximated through first-order Taylor series expansion, neglecting higher-order terms, resulting in the following:(16)$$e_{i}^{\left(\varphi \right)}\approx e_{i}^{\left(\varphi \right)}\left(x^{\left(k\right)}\right)+\frac{y-a_{iy}}{{\left(x-a_{ix}\right)}^{2}+{\left(y-a_{iy}\right)}^{2}}\Delta x-\frac{x-a_{ix}}{{\left(x-a_{ix}\right)}^{2}+{\left(y-a_{iy}\right)}^{2}}\Delta y$$(17)$$e_{i}^{\left(\tau \right)}\approx e_{i}^{\left(\tau \right)}\left(x^{\left(k\right)}\right)+\frac{x-a_{ix}}{c\left\|x^{\left(k\right)}-a_{i}\right\|}\Delta x+\frac{y-a_{iy}}{c\left\|x^{\left(k\right)}-a_{i}\right\|}\Delta y$$(18)$$e_{i}^{\left(d\tau \right)}\approx e_{i}^{\left(d\tau \right)}\left(x^{\left(k\right)}\right)-\frac{1}{c}\left(\frac{x-a_{ix}}{\left\|x^{\left(k\right)}-a_{i}\right\|}-\frac{x-a_{1x}}{\left\|x^{\left(k\right)}-a_{1}\right\|}\right)\Delta x-\frac{1}{c}\left(\frac{y-a_{iy}}{\left\|x^{\left(k\right)}-a_{i}\right\|}-\frac{y-a_{1y}}{\left\|x^{\left(k\right)}-a_{1}\right\|}\right)\Delta y$$

The gradient expressions $J_{i}^{\left(\varphi \right)}$, $J_{i}^{\left(\tau \right)}$, and $J_{i}^{\left(d\tau \right)}$ are derived from Equations (16)–(18):(19)$$J_{i}^{\left(\varphi \right)}=\left[\begin{array}{cc} \frac{y^{\left(k\right)}-a_{iy}}{{\left(x^{\left(k\right)}-a_{ix}\right)}^{2}+{\left(y^{\left(k\right)}-a_{iy}\right)}^{2}} & -\frac{x^{\left(k\right)}-a_{ix}}{{\left(x^{\left(k\right)}-a_{ix}\right)}^{2}+{\left(y^{\left(k\right)}-a_{iy}\right)}^{2}} \end{array}\right]$$(20)$$J_{i}^{\left(\tau \right)}=\left[\begin{array}{cc} \frac{x^{\left(k\right)}-a_{ix}}{c\left\|x^{\left(k\right)}-a_{i}\right\|} & \frac{y^{\left(k\right)}-a_{iy}}{c\left\|x^{\left(k\right)}-a_{i}\right\|} \end{array}\right]$$(21)$$J_{i}^{\left(d\tau \right)}=\left[\begin{array}{cc} -\frac{x^{\left(k\right)}-a_{ix}}{c\left\|x^{\left(k\right)}-a_{i}\right\|}+\frac{x^{\left(k\right)}-a_{1x}}{c\left\|x^{\left(k\right)}-a_{1}\right\|} & -\frac{y^{\left(k\right)}-a_{iy}}{c\left\|x^{\left(k\right)}-a_{i}\right\|}+\frac{y^{\left(k\right)}-a_{1y}}{c\left\|x^{\left(k\right)}-a_{1}\right\|} \end{array}\right]$$

For each set of AOA, TOA, and TDOA measurements, the linear equations can be rearranged into the following form:(22)$$e_{i}\approx J_{i}\Delta x+\epsilon_{i}$$

Here, $\Delta x={\left[\Delta x,\Delta y\right]}^{\;T}$, $J_{i}$ represents the gradient matrix, and $\epsilon_{i}$ denotes the residual term of the equation. By rearranging Equation (22), the matrix form can be written as follows:(23)JΔx=r
where J represents the Jacob matrix, and r denotes the residual error vector. The expression of J under the four positioning modes is given as follows:(24)$$\left\{\begin{array}{l} J^{\left(AOA\right)}=\left[J_{1}^{\left(AOA\right)},J_{2}^{\left(AOA\right)},\ldots ,J_{N_{gs}}^{\left(AOA\right)}\right]\\ J^{\left(TOA\right)}=\left[J_{1}^{\left(TOA\right)},J_{2}^{\left(TOA\right)},\ldots ,J_{N_{gs}}^{\left(TOA\right)}\right]\\ J^{\left(AOA/TOA\right)}=\left[J_{1}^{\left(AOA\right)},J_{2}^{\left(AOA\right)},\ldots ,J_{N_{gs}}^{\left(AOA\right)},J_{1}^{\left(TOA\right)},J_{2}^{\left(TOA\right)},\ldots ,J_{N_{gs}}^{\left(TOA\right)}\right]\\ J^{\left(AOA/TDOA\right)}=\left[J_{1}^{\left(AOA\right)},J_{2}^{\left(AOA\right)},\ldots ,J_{N_{gs}}^{\left(AOA\right)},J_{1}^{\left(TDOA\right)},J_{2}^{\left(TDOA\right)},\ldots ,J_{N_{gs}-1}^{\left(TDOA\right)}\right] \end{array}\right.$$

Large residuals in the positioning equations, resulting from NLOS noise, adversely affect solution accuracy. NLOS noise can be effectively suppressed through robust loss functions that respond to residual magnitudes. Denoting the robust loss function as $\rho \left(e\right)$ with its first derivative $\psi \left(e\right)$, the iterative weight update can be expressed as follows:(25)$$w_{iter,i}^{\left(k\right)}=\frac{\psi \left(e_{i}^{\left(k\right)}\right)}{e_{i}^{\left(k\right)}}$$

Hence, the iterative weight matrix determined by the residuals is given by the following:(26)$$W_{iter}^{\left(k\right)}=diag\left(w_{iter,1}^{\left(k\right)},w_{iter,2}^{\left(k\right)},\ldots ,w_{iter,N_{gs}}^{\left(k\right)}\right)$$

The representative robust loss functions commonly employed include Huber loss, Cauchy loss, Tukey loss, and Geman–McClure loss as follows:(27)$$\left\{\begin{array}{l} \rho_{Huber}\left(e\right)=\left\{\begin{array}{ll} 0.5e^{2} & \mathrm{if}\left|e\right|\leq \delta ,\\ \delta \left(\left|e\right|-0.5\delta \right) & \mathrm{otherwise} \end{array},\right.\\ \rho_{Cauchy}\left(e\right)=0.5\delta^{2}\ln\left(1+e^{2}/\delta^{2}\right),\\ \rho_{Tukey}\left(e\right)=\left\{\begin{array}{ll} \delta^{2}\left(1-{\left(1-e^{2}/\delta^{2}\right)}^{3}\right) & \mathrm{if}\left|e\right|\leq \delta ,\\ \delta^{2} & \mathrm{otherwise} \end{array}.\right.\\ \rho_{Geman-McClure}\left(e\right)=e^{2}/\left(\delta^{2}+e^{2}\right). \end{array}\right.$$

The damping parameter δ for each iteration is defined as the standard deviation of all residuals. This threshold effectively reduces the weights of GSs exhibiting large residual deviations, thereby accelerating convergence and enhancing positioning efficiency. The IRLS method implementation employs two termination criteria: a maximum iteration count of 30 and a position increment threshold of $\left\|\Delta x\right\|<1\times 10^{-5}$.

Thus, the total weight matrix at k-th iteration $W^{\left(k\right)}$ is given by the following:(28)$$W^{\left(k\right)}=W_{0}⊙W_{iter}^{\left(k\right)}$$
where $W_{0}$ is the initialized weight matrix, and ⊙ denotes the Hadamard product. Since both $W_{0}$ and $W_{iter}^{\left(k\right)}$ are diagonal matrices, their element-wise product remains a diagonal matrix. Specifically, its diagonal elements can be expressed as follows:(29)$$w_{i}^{\left(k\right)}=w_{0,i}\cdot w_{\mathrm{iter},i}^{\left(k\right)},i=1,2,\ldots ,N_{gs}$$
where $w_{i}^{\left(k\right)}$, $w_{0,i}$, and $w_{\mathrm{iter},i}^{\left(k\right)}$ are the i-th diagonal elements of $W^{\left(k\right)}$, $W_{0}$, and $W_{iter}^{\left(k\right)}$, respectively.

With the weight matrix $W^{\left(k\right)}$ introduced, the position increment Δx is derived via the IRLS formulation as follows:(30)$$\Delta x^{\left(k\right)}={\left(J^{T}W^{\left(k\right)}J\right)}^{-1}J^{T}W^{\left(k\right)}c$$

The position estimate for the $\left(k+1\right)$-th iteration is calculated as follows:(31)$$x^{\left(k+1\right)}=x^{\left(k\right)}+\Delta x^{\left(k\right)}$$

## 4. Experimental Results in Underground Parking Garage

### 4.1. Measurement Equipment

A dual-channel UWB direction-finding system was deployed for this measurement campaign, as illustrated in Figure 4. The system integrated a DMW1000 chip for UWB signal measurements with an NXP SR150 MCU, which processed the dual-channel UWB data and computed AOA estimates [61]. The data were transmitted to a local host through a UART interface. Performance testing demonstrated the UWB device achieved measurement accuracies of 0.1 m for time delay and 8° for angle determination.

### 4.2. Measurement Scenario

The experiments were conducted in a relatively open underground parking garage. A handheld laser LiDAR system, with 5 cm modeling accuracy, was employed to capture the scene’s geometric structure. Figure 5 displays the raw point cloud data of the garage, revealing multiple wall structures and isolated rooms that create numerous NLOS conditions. Figure 6 shows the geometric model extracted from the point cloud, encompassing a 50 m × 50 m area. The test environment features two major room obstructions and several load-bearing columns.

Figure 7 shows the point cloud collection device, which incorporates an IMU module for precise position measurement. The device’s integrated SLAM module provided accurate position determination, eliminating reference frame discrepancy errors. The UWB tag was mounted on top of the point cloud device.

The measurement setup comprised four UWB anchors and one UWB tag. The anchors were strategically mounted on walls throughout the scene shown in Figure 8, at a consistent height of 1.85 m above the ground, at coordinates $UWB_{0}={\left[22.47,33.97\right]}^{\;T}$, $UWB_{1}={\left[23.73,16.58\right]}^{\;T}$, $UWB_{2}={\left[38.84,26.75\right]}^{\;T}$, and $UWB_{3}={\left[40.0,8.77\right]}^{\;T}$. Portable power banks supplied power to each anchor. To establish accurate reference path coordinates for evaluation, UWB parameter collection was conducted simultaneously with point cloud scanning.

### 4.3. Localization Accuracy Validation

This section analyzes the UWB device data and validates the positioning algorithm’s accuracy. The measurement trajectory shown in Figure 7 was completed in approximately 160 s at a constant speed of 1.5 m/s. Figure 9a depicts the UWB channel propagation distances over time, showing time delay variations due to multipath effects, which are particularly significant under NLOS conditions. Figure 9b,c present the UWB-measured angle and power data.

Five position estimators were compared to validate the algorithm’s accuracy, all utilizing the GS technique.

(1)W-IRLS (proposed algorithm): Incorporates the initial weighted matrix $W_{0}$ and uses the optimal GSPC as the initial solution.(2)IRLS: Uses the optimal GSPC as the initial solution but assigns equal weights to all equations.(3)TSWLS: Implements the classical two-step weighted least squares approach, using only the LS method for the initial solution without initial weights.(4)WLS: Employs both the weighted matrix $W_{0}$ and the optimal GSPC as the initial solution.(5)LS: Directly solves equations using the least squares approach, without weights or initial solution.

The evaluation framework encompasses four positioning methods based on UWB device measurements: AOA-based positioning, TOA-based positioning, hybrid AOA-TOA positioning, and hybrid AOA-TDOA positioning.

Figure 10 illustrates the comparison of these methods’ positioning results with the actual trajectory. Figure 11 displays the cumulative distribution functions (CDFs) for each positioning algorithm. The positioning errors of different algorithms were analyzed across four positioning modes: AOA, TOA, AOA/TOA, and AOA/TDOA. The average location error (ALE) and standard deviations (STDs) for each mode are summarized in Table 1 and Table 2, respectively. While all five algorithms exhibit suboptimal performance in the AOA positioning mode due to significant measurement errors, the proposed W-IRLS method demonstrates superior performance with an accuracy of 2.17 m. The W-IRLS method continues to outperform in other positioning modes, achieving average accuracies of 0.18 m in TOA mode, 0.14 m in AOA/TOA mode, and 0.3 m in AOA/TDOA mode. The consistently superior performance of the W-IRLS method across all positioning modes underscores the effectiveness of our proposed weighted matrix and optimal initialization based on the GS technique.

## 5. Robust Analysis of RT-VLBS Framework

The GS technique achieves high positioning accuracy in NLOS scenarios, but its performance depends on measurement precision and geometric modeling accuracy. While UWB platforms provide high-precision time delay measurements, extreme conditions can increase measurement errors, challenging algorithm noise robustness. In terms of geometric modeling, point cloud modeling offers high precision but limited practical applicability, while common maps like Planet and OpenStreetMap have decimeter-to-meter-level accuracy, impacting GS positioning algorithms. This section examines how measurement errors and geometric modeling precision affect positioning algorithm performance.

### 5.1. Simulation Environment

Figure 12 depicts a 50 m × 50 m NLOS scenario with smooth-plane reflector surfaces. Anchor and tag positions were selected to test diverse propagation conditions. Three anchors were positioned at $x_{B1}\left(10,48\right)$, $x_{B2}\left(9,21\right)$, and $x_{B3}\left(20,22\right)$, while tags were placed at three locations: $A\left(14,31\right)$, $B\left(25,39\right)$, and $C\left(38,28\right)$. Position A provides LOS conditions between the tag and all anchors. At position B, only $x_{B1}$ maintains LOS with the tag, while other anchor paths involve single reflections or diffractions. Position C represents complete NLOS conditions, where UWB signals reach anchors through multiple reflections and diffractions.

### 5.2. Comparison of Localization Accuracy with Different AOA Errors

The accuracy and robustness of the proposed W-IRLS algorithm based on the GS technique were evaluated through two simulation rounds. The first round assessed positioning accuracy under varying error conditions. In the first simulations, TOA error followed a Gaussian distribution with 3 ns STD, while RSS error was modeled as Gaussian with 6 dB STD, reflecting typical RT model accuracy. AOA error was simulated as a Gaussian random variable with an STD ranging from 0.1° to 8°. Root mean square error (RMSE) calculations were based on 10,000 independent simulation runs.

Figure 13 demonstrate the W-IRLS algorithm’s positioning accuracy under varying angular error conditions. While increasing AOA errors leads to higher positioning errors in AOA-based algorithms, methods incorporating time-based parameters (TOA and TDOA) achieve notably superior accuracy with sub-meter-level precision. Table 3 and Table 4 show the statistics of ALEs of different methods. The positioning method incorporating the GS technique proves both accurate and robust in LOS and NLOS conditions.

### 5.3. Comparison of Localization Accuracy Under Different Map Errors

The second simulation round examined positioning accuracy under varying map errors, with RMSE calculated from 10,000 independent runs. The simulation used zero-mean Gaussian distributions for TOA (3 ns STD), AOA (2° STD), and RSS (6 dB STD) errors. Figure 14 shows the geometric building corner point p error, modeled as a zero-mean Gaussian distribution with a STD $\sigma_{Displacement}$ ranging from 0.01 m to 2.0 m. Corner position displacement affects wall tilt angles, altering GS locations and consequently introducing positioning system errors.

Figure 15a illustrates positioning errors for the tag at point A across different modes. Environmental displacement shows minimal impact on all four algorithms’ accuracy. This stability stems from point A’s LOS conditions with all three anchors, where the absence of reflection or diffraction paths renders the positioning independent of environmental modeling errors.

Figure 15b shows positioning errors for the tag at point B across different modes. Algorithm accuracy degrades with increasing environmental displacement. While the tag maintains LOS conditions with $x_{B_{1}}$, its reflection paths to $x_{B_{2}}$ and $x_{B_{3}}$ are dependent on environmental modeling accuracy, thus affecting overall positioning performance.

Figure 15c demonstrates positioning errors for the tag at point C across different modes. The positioning accuracy shows the highest sensitivity to environmental modeling errors, as all paths between the tag and anchors are NLOS. This increased sensitivity is reflected in the error curve’s steeper slope compared to point B’s results.

Table 5 summarizes the errors caused by average geometric modeling accuracy for four different positioning modes. From the perspective of the tag’s position, the complexity of the channel between the tag and the anchors (e.g., the number of multipath bounces) directly affects the accuracy of the positioning algorithms. In other words, the more complex the multipath, the greater the error caused by geometric modeling inaccuracies.

Analysis of the four positioning modes reveals distinct performance characteristics: The AOA/TOA algorithm demonstrates superior robustness, achieving a positioning error of 0.83 m per unit of geometric modeling accuracy in fully NLOS scenarios. Following closely, the TOA algorithm exhibits strong performance with a 0.98 m error per unit. The AOA/TDOA algorithm ranks third with a 1.67 m error per unit, while the AOA/TDOA positioning algorithm shows the highest environmental sensitivity, resulting in a 2.0 m error per unit of geometric modeling accuracy.

## 6. Discussion and Future Work

Extensive experiments and simulations conducted in underground parking garages demonstrate that the proposed RT-VLBS platform performs exceptionally well in NLOS scenarios. However, we outline two limitations of the proposed algorithm and identify key issues to address in future work:(1)Dynamic Environments: The proposed algorithm was validated in a completely static underground parking garage. It does not account for the dynamic characteristics of parking garages, such as the presence of pedestrians and vehicles, which can introduce additional power attenuation and delay to UWB signals. Future work should evaluate the impact of dynamic factors (e.g., pedestrians and vehicles) on the positioning performance of RT-VLBS. Statistical analysis methods should be employed to establish relationships between dynamic features and UWB signal propagation characteristics, improving the adaptability of the RT-VLBS method.(2)2.5D Limitations: The proposed RT-VLBS method operates in a 2.5D framework. However, in scenarios with sloped surfaces or spiral ramps, as commonly found in underground parking garages, more complex 3D structural features must be taken into account. Future work should focus on developing a fully 3D RT-VLBS method to enhance positioning accuracy in these challenging environments.

Furthermore, we suggest the following areas as key focal points for future research:(1)Anchor Placement Optimization: Traditional anchor placement is typically assessed based on the geometric dilution of precision (GDOP) in LOS scenarios. However, RT-VLBS leverages multipath signals for positioning. Therefore, future research should explore anchor placement strategies that consider the GDOP in the context of the RT-VLBS method, optimizing placement to improve positioning accuracy.(2)Integration with Intelligent Reflecting Surfaces: The emergence of intelligent reflecting surfaces (IRS) offers the potential to alter the propagation direction of electromagnetic waves, introducing new channel information. In the planning of future smart parking garages, integrating the RT-VLBS method with IRS could significantly enhance positioning performance.(3)Quasi-Specular reflection modeling: As 6G adopts higher-frequency signals, millimeter-wave propagation becomes increasingly sensitive to wall surface irregularities, deviating from ideal specular reflection. Thus, future research should integrate quasi-specular reflection models into localization algorithms to enhance adaptability in the millimeter-wave regime.

These proposed directions aim to address the identified limitations and further improve the RT-VLBS platform’s robustness and applicability in practical scenarios.

## 7. Conclusions

This paper proposes an innovative UWB positioning framework for NLOS scenarios based on the GS technique. In complex NLOS environments, the RT algorithm converts NLOS paths into equivalent LOS paths. A state-of-the-art GS filtering and weighting method is designed to determine the initial weights for solving nonlinear equations. Furthermore, we propose a robust W-IRLS method that effectively mitigates the adverse effects of complex NLOS noise environments, substantially improving the positioning algorithm’s resilience and reliability. The framework comprehensively evaluates four distinct positioning modes: AOA, TOA, hybrid AOA/TOA, and integrated AOA/TDOA. Comprehensive experimental validation and simulation studies demonstrate the superior performance of our proposed NLOS positioning algorithm, achieving an exceptional positioning accuracy of 0.14 m in challenging NLOS environments. This breakthrough research establishes a robust theoretical and practical foundation for next-generation high-precision NLOS positioning services, facilitating the seamless integration of communication and sensing technologies.

## Figures and Tables

**Figure 1 sensors-25-02082-f001:**
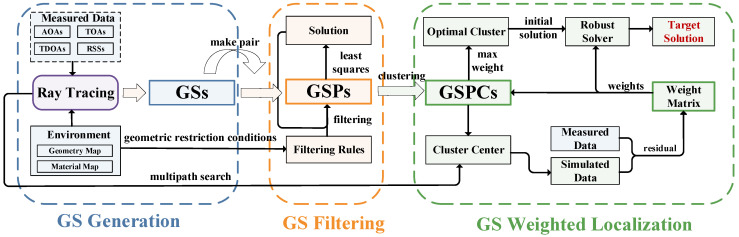
Flowchart of RT-VLBS algorithm. The algorithm consists of three main components: (1) GS generation based on ray-tracing and environmental data, (2) GS filtering through geometry restriction rules, and (3) GS weighted localization incorporating robust solver and weight matrix.

**Figure 2 sensors-25-02082-f002:**
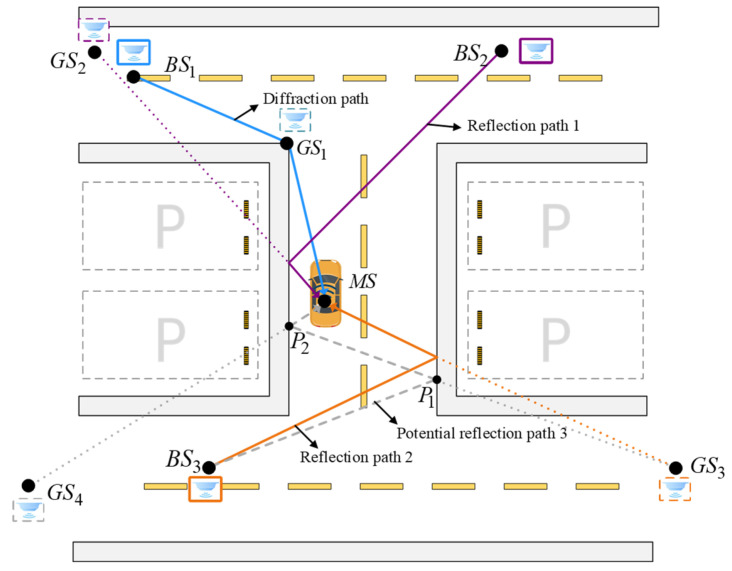
Vehicle localization based on GSs from multipath propagation in an underground parking scenario. The figure illustrates the diffraction path and multiple reflection paths between base stations (BSs) and mobile stations (MSs), with corresponding GSs generated from different propagation mechanisms. The dotted lines represent backward extensions for GS searching.

**Figure 3 sensors-25-02082-f003:**
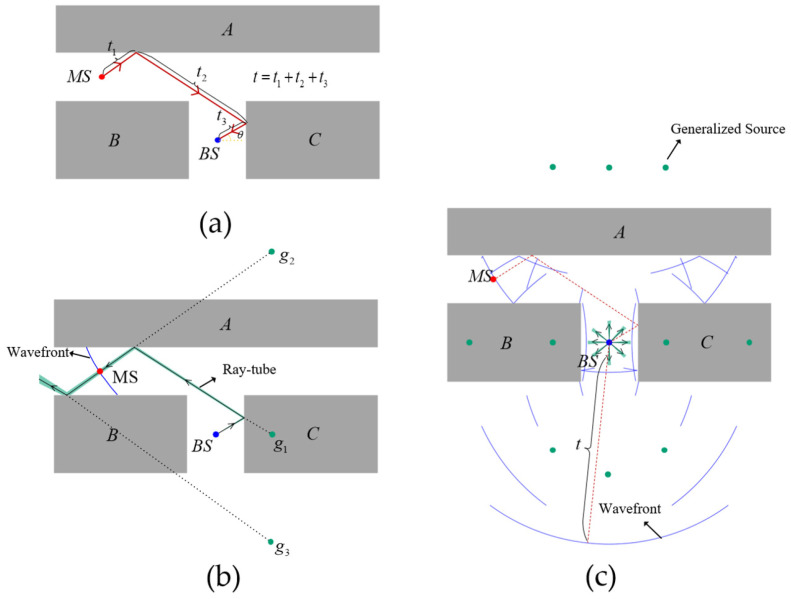
Schematic diagram of GS generation. (**a**) Signal propagation path between the mobile station (*T*) and the base station (*R*). (**b**) Schematic diagram of GS generation in the localization mode with angle information. (**c**) Schematic diagram of GS generation in the TOA localization mode. A, B, and C are obstacles. The dotted lines represent backward extensions for GS searching.

**Figure 4 sensors-25-02082-f004:**
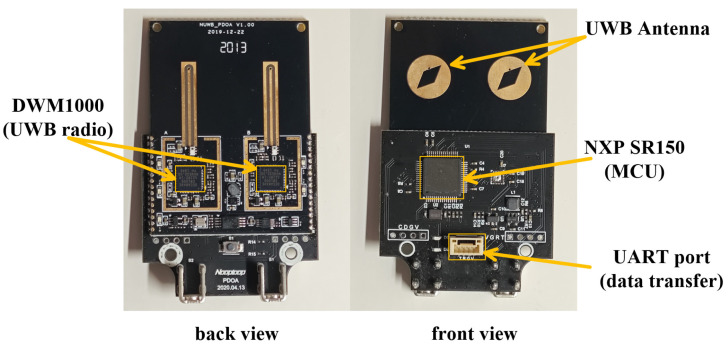
Front and back views of the dual-channel UWB receiver module. The back view (**left**) shows the DWM1000 UWB radio modules, while the front view (**right**) features UWB antennas, an NXP SR150 microcontroller (MCU), and a UART port for data transfer.

**Figure 5 sensors-25-02082-f005:**
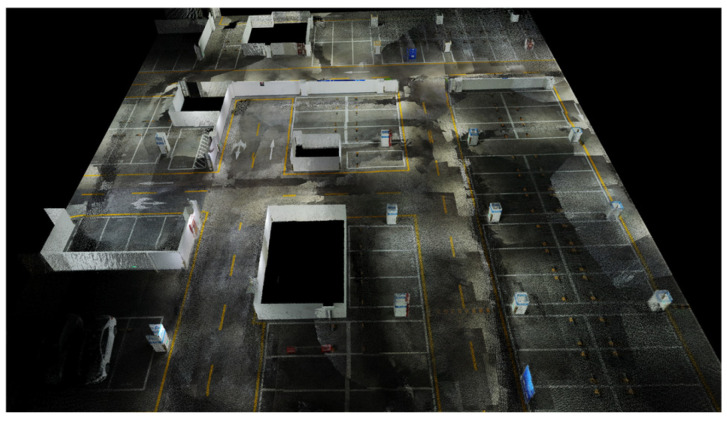
Point cloud map of the underground parking garage.

**Figure 6 sensors-25-02082-f006:**
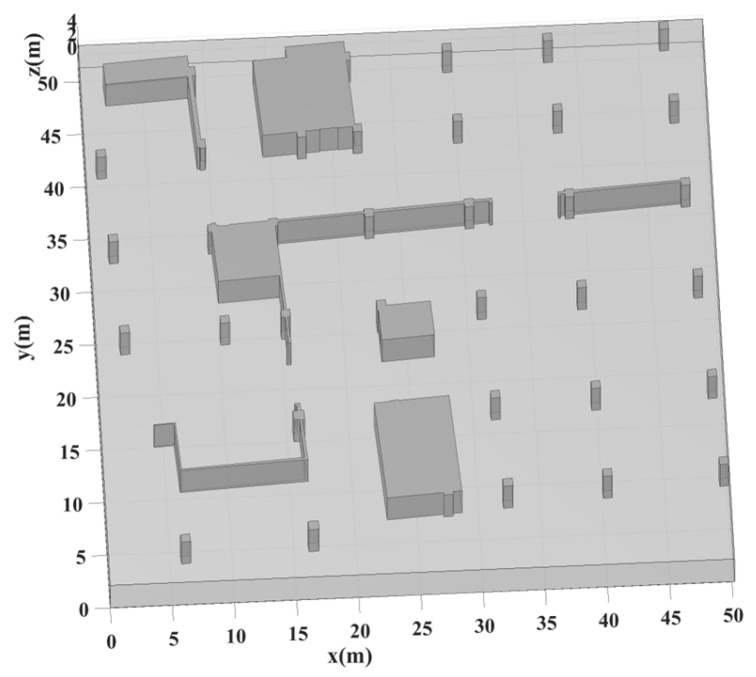
Geometric model of the underground parking garage., where gray structures represent walls and columns.

**Figure 7 sensors-25-02082-f007:**
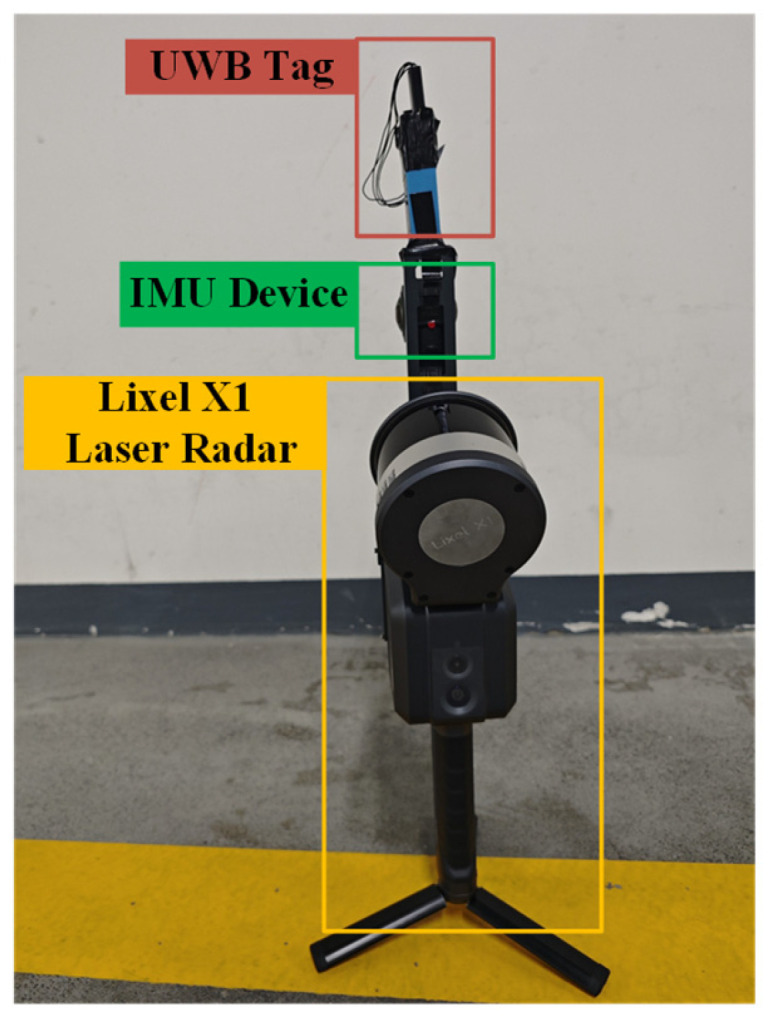
The integrated system of UWB-Tag and point cloud scanning equipment. The system consists of a UWB tag, inertial measurement unit (IMU) device, and Lixel X1 LiDAR (Hi-Target Navigation Tech Co. Ltd., Guangzhou, China), enabling high-precision 3D positioning and mapping. The scanning equipment provides global coordinate reference while the IMU assists UWB-tag in pose estimation.

**Figure 8 sensors-25-02082-f008:**
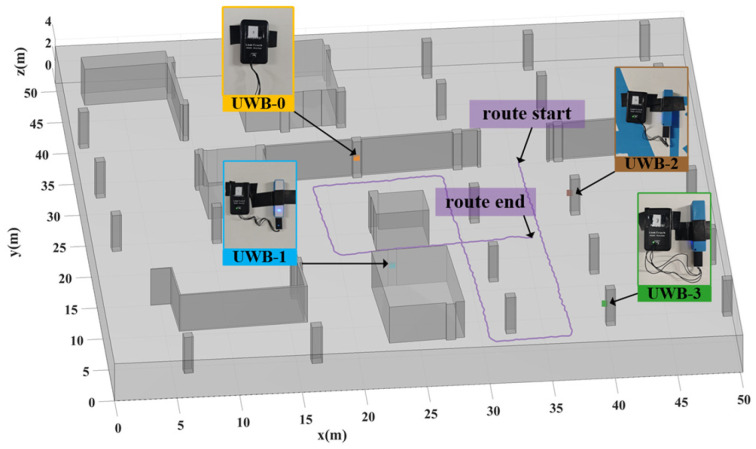
Layout of UWB anchors and tag trajectory. The figure shows the spatial deployment of four UWB anchors (UWB-0 to UWB-3) in the underground parking garage, where gray structures represent walls and columns. The actual trajectory of the UWB tag is also illustrated (from route start to route end).

**Figure 9 sensors-25-02082-f009:**
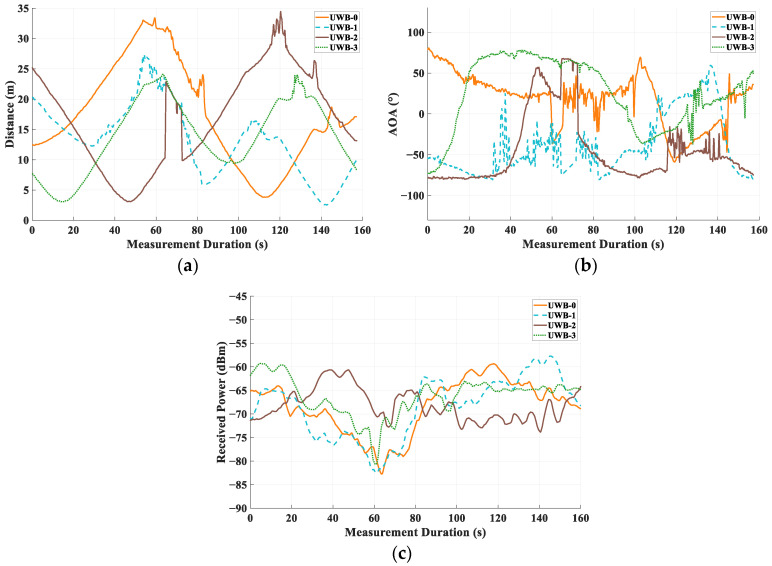
UWB-measured data. (**a**) Distance measurements of UWB anchors. (**b**) Distance measurements of UWB anchors. (**c**) Received power measurements of UWB anchors.

**Figure 10 sensors-25-02082-f010:**
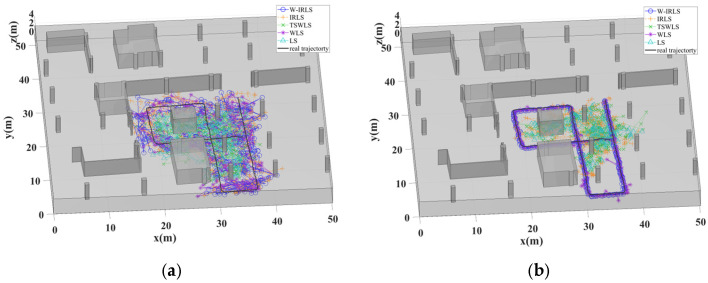
Comparison of different positioning algorithms with the real trajectory. (**a**) In AOA localization mode. (**b**) In TOA localization mode. (**c**) In AOA/TOA localization mode. (**d**) In AOA/TDOA localization mode.

**Figure 11 sensors-25-02082-f011:**
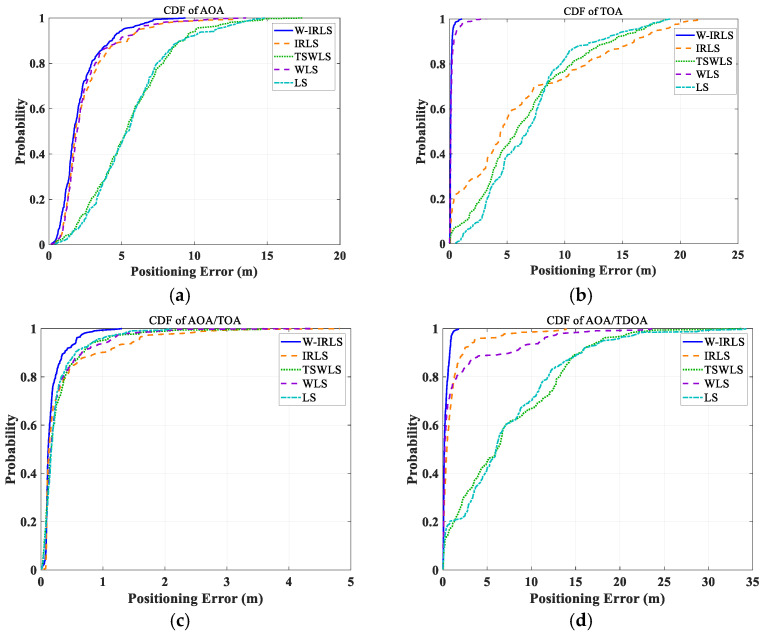
The CDF of positioning error for different algorithms. (**a**) In AOA localization mode (with $\sigma_{\varphi }=8^{\circ }$). (**b**) In TOA localization mode (with $\;\sigma_{\tau }=0.1\;m$). (**c**) In AOA/TOA localization mode (with $\sigma_{\varphi }=8^{\circ },\;\;\sigma_{\tau }=0.1\;m$). (**d**) In AOA/TDOA localization mode (with $\sigma_{\varphi }=8^{\circ },\sigma_{d\tau }=0.1\;m$).

**Figure 12 sensors-25-02082-f012:**
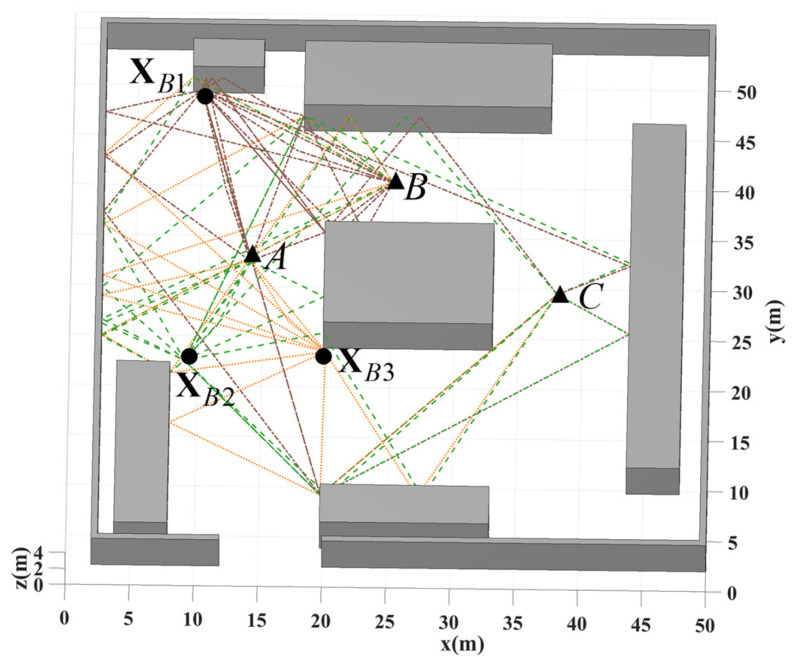
Simulation environment with horizontal and vertical reflectors. $x_{B1}$, $x_{B2}$, and $x_{B3}$ represent anchor positions, with tags placed at points A, B, and C. The colored lines represent multipath propagation between different anchor-tag pairs.

**Figure 13 sensors-25-02082-f013:**
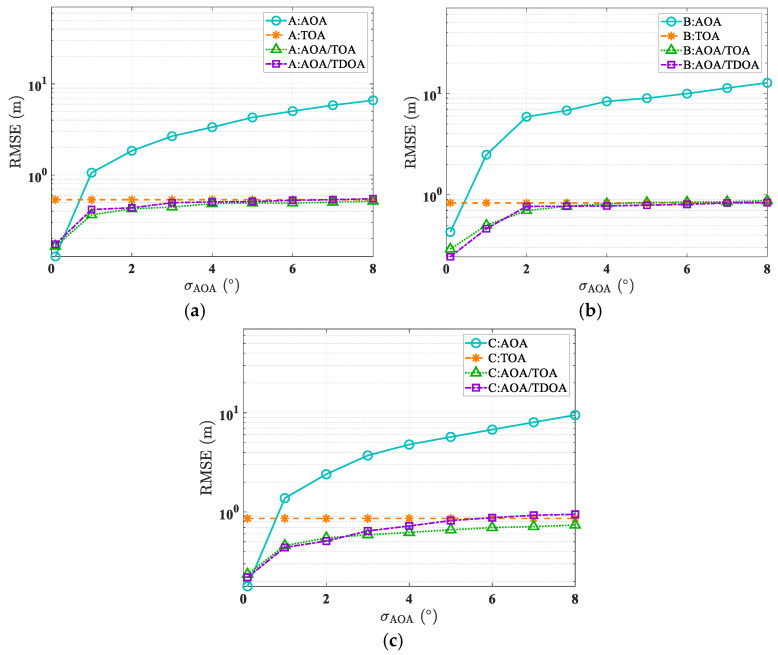
RMSE performance of different scenarios under different localization modes and varying angle error conditions. (**a**) In scenario A. (**b**) In scenario B. (**c**) In scenario C.

**Figure 14 sensors-25-02082-f014:**
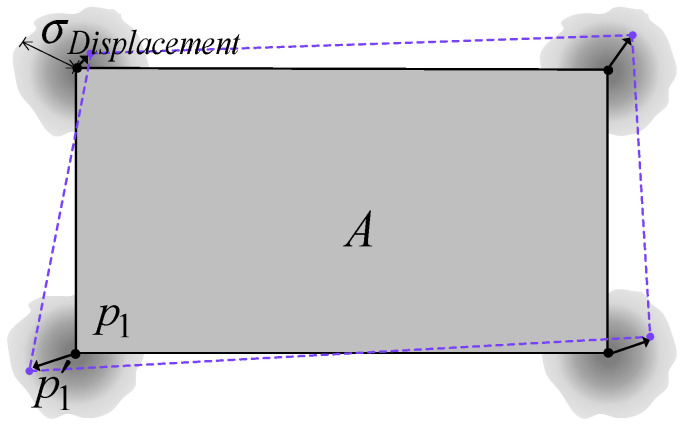
Schematic diagram of building corner position error. The displacement distance $\sigma_{Displacement}$ follows zero-mean Gaussian distribution, and $p_{1}^{\prime }$ is the perturbed coordinate of $p_{1}$. The gray shaded areas represent the possible locations of building corners due to random displacement errors. The dashed lines indicate the perturbed geometry of the building after considering these uncertainties.

**Figure 15 sensors-25-02082-f015:**
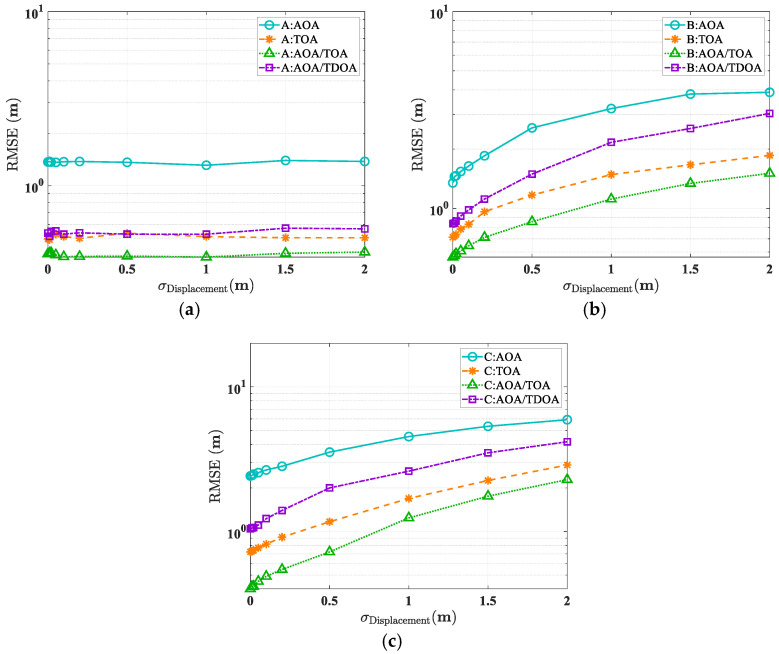
RMSE performance of different scenarios under different localization modes and varying geometric model error conditions. (**a**) In scenario A. (**b**) In scenario B. (**c**) In scenario C.

**Table 1 sensors-25-02082-t001:** Mean of different algorithms’ ALEs under various localization modes.

Algorithm	W-IRLS	IRLS	TSWLS	WLS	LS
AOA	2.17 m	2.56 m	5.53 m	2.48 m	5.62 m
TOA	0.18 m	6.30 m	6.71 m	0.21 m	6.96 m
AOA/TOA	0.14 m	0.35 m	0.29 m	0.29 m	0.27 m
AOA/TDOA	0.30 m	1.07 m	7.25 m	1.77 m	7.23 m

**Table 2 sensors-25-02082-t002:** Standard deviation of different algorithms’ ALEs under various localization modes.

Algorithm	W-IRLS	IRLS	TSWLS	WLS	LS
AOA	1.48 m	1.97 m	2.86 m	1.83 m	2.81 m
TOA	0.17 m	5.85 m	4.68 m	0.32 m	4.06 m
AOA/TOA	0.12 m	0.56 m	0.41 m	0.41 m	0.31 m
AOA/TDOA	0.31 m	1.97 m	6.19 m	3.86 m	6.26 m

**Table 3 sensors-25-02082-t003:** Mean of different localization modes’ ALEs.

Algorithm	AOA	TOA	AOA/TOA	AOA/TDOA
A	3.23 m	0.54 m	0.43 m	0.46 m
B	7.44 m	0.83 m	0.69 m	0.72 m
C	4.72 m	0.86 m	0.58 m	0.68 m

**Table 4 sensors-25-02082-t004:** Standard deviation of different localization modes’ ALEs.

Algorithm	AOA	TOA	AOA/TOA	AOA/TDOA
A	3.65 m	0.33 m	0.27 m	0.29 m
B	8.69 m	0.48 m	0.46 m	0.55 m
C	9.21 m	0.58 m	0.49 m	0.51 m

**Table 5 sensors-25-02082-t005:** Mean of different localization modes’ ALEs under unit building displacement error.

Algorithm	AOA	TOA	AOA/TOA	AOA/TDOA
A	0 m	0 m	0 m	0 m
B	1.8 m	0.72 m	0.52 m	1.22 m
C	2.0 m	0.98 m	0.83 m	1.67 m

## Data Availability

The data that support the findings of this study are available by contacting the corresponding author.

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
