# Peer review of "Multipath-Assisted Ultra-Wideband Vehicle Localization in Underground Parking Environment Using Ray-Tracing"

_sensors, 2025, doi:10.3390/s25072082_

Round 1

Reviewer 1 Report

Comments and Suggestions for Authors

This paper proposes a Non-Line-of-Sight (NLOS) positioning framework based on Generalized Source (GS) technology, which transforms NLOS paths into equivalent Line-of-Sight (LOS) paths. A novel GS filtering and weighting strategy is introduced to establish initial weights for the nonlinear equation system, and an Iteratively Reweighted Least Squares (W-IRLS) method is employed to integrate the initial weights with optimal position estimation. By leveraging Ultra-Wideband (UWB) delay and angle measurements, four distinct positioning modes based on W-IRLS are developed, effectively mitigating the impact of NLOS on precise vehicle localization in complex underground parking scenarios.

 This paper needs to be supplemented and improved, as described below.

1 All elements illustrated in Figure 1 should be depicted within the context of this paper, such as the measured data, environment, and Ray Tracing in the GS generation process.

2 In the paragraph above Algorithm 1, "For instance, as depicted in Figure 2, …", the elements R1 and R2 should be explicitly labeled in Figure 2. How can we determine whether obstacles exist in R_MS1 and R_MS2? Furthermore, it should be clarified whether GS3 and GS4 exhibit geometric validity in the example.

3 In Algorithm 1, "Pairing the GSs to construct CN2 GSPs," the method used for pairing, such as random or other approaches, should be explicitly described.

4 Is it guaranteed that the number of GSPs will not be zero after executing Algorithm 1? If so, what is the reason for this assurance?

5 In Equation (4), is Ms a vector, an array, or a set? Is Ms independent of the index i?

6 The specific steps for applying the max() function in Equations 5, 6, 7, 8, and 9 should be explained in detail.

7 In Equation 4, N represents the number of BSs, but in the description below Equation 9, N denotes the total number of GSs, which may confuse readers.

8 All parameters used in the equations of this paper should be clearly and thoroughly defined, such as W0, W0,ii, w0,i, and others.

Reviewer 2 Report

Comments and Suggestions for Authors

Authors asses UWB based Localization in the underground environment. Authors could make correction to the manuscript if I missed some points. Hence 2D case is a good assumption, and authros refers it as 2.5D, which I understand as z (height) variations is much smaller than x and y variations. In my opinion the approach is in its spirit close to SLAM which asses localization and mapping simultaneously. (Point 1) Here authors also generated GS based on measurements BUT also based on current object location. I assume GS generation is continuous since object moves. If it is not the case please correct, then the Figure 2 doesn’t reflect the geometry of experiment. In the case objects’ position is part of the iterative process of localization, there should convergence interval during which there is steady error descending. I could not see the data that reflect the convergence. From Figure 13 it is clear that error is small from the very beginning of the experiment. Is initial position setup is pre-requirements for method to work? (Point 2) Authors uses four modes: AOA, TOA, AOA/TOA, AOA/TDOA, and if we refer Figure 2 we should notice that GS could only be placed as they are placed only if both time delay and angle is known. Is this Figure demonstrate only some modes? If it demonstrates all mode, could you please put additional description how it refers to single AOA and single TOA modes. (Point 3) I didn’t get from the manuscript why GS1 (diffraction) should be different from all other GSs. Why it is placed on the corner of the obstacles i.e.  why time delay is not counted in case of GS1? Please add description of how diffraction affects GS placement. (Point 4) Authors honestly write that dynamic environment is the objective of future work, however I could notice  potential issue that authors should also reflect in the discussion: the geometry of the wall. While material that influence the absorption of a signal should not affect authors method much ( since the AOA and TOA used) the objects or wall geometry itself can break assumption angle-of-incidence-is-equal-to-angle-of-reflection, which is used according to Figure2.

I believe that manuscript benefit from combining similar Figure example (11-14 or 15-18) into single Figure with four panels.

Since line numbering is not given I could only put text that I think have typos

(Page 3) a two‐step least squares (TSWLS) -> a two‐step weighted least squares (TSWLS)

(Page 6) with corresponding reflection points  R1  and  R2 - > P1  and  P2

Round 2

Reviewer 1 Report

Comments and Suggestions for Authors

No further comments.

Reviewer 2 Report

Comments and Suggestions for Authors

Dear Authors, thank you for the detailed response to my comment.

Response 1:

Thank you for detailed response.

Response 2:

Thank you for clear response.

Form the response I can learn that geometry of the environment is known: “According to the assumptions in our manuscript, the positions of the base station (BS) and the geometric environment are fixed. Under this condition, the GS position generated by the base station remains fixed.” I initially missed this requirement (known geometry) of the proposed method. I think it would be convenient for a reader to state it explicitly. Since it reads (at least for me) like geometry is reconstructed during the operation.

It is also confusing that according to the Figure R1 base station receives signal while according to the Figures in the manuscript a base station emits signal that is received by mobile station.

Response 3:

Thank you for additional materials, it demonstrates authors willing to improve the manuscript.

Additional Figure (Figure 3 in second version of the manuscript) clarifies GS generation. In my opinion, it is confusing that base station and mobile station a flipped in Figure 2 and 3: mobile station (𝑇 ) and the base station (𝑅) in Figure 3 are at the locations of base station BS1 and mobile station (MS) in the Figure 2 correspondingly. It didn’t limit my understanding of concepts author illustrated, however it makes a reader to go back and compare the notations. Different notation is also confusing, why don’t keep going with MS and BS?

Response 4:

Thank you for the response. However, I still miss few points here.

Again, it is confusing that base station receives signal while according to the Figures in the manuscript a base station emits signal that is received by mobile station.

Huygens–Fresnel principle states that every point of wave-front is itself secondary source and hence we can see signal in radio shadow as you pointed. I’m not sure why the diffraction breaks AOA mode  (if I understand correctly the statement “diffracted GS does not have a well‐defined AOA due to the nature of diffraction”). According to Figure R3 the wavefront is spherical and in that sense is not different from circular arc drawn for reflected signal. The response also assumes that time-of-arrival is measured correctly. I certainly understand that there could be some physical or technical issues that prevent usage of angle-of-arrival, I hope authors could point it in the manuscript for readers like me.

Anyway, I try to rephrase the comment in the notation provided by Figure R3. How t2 (RD distance) is utilized when constructed GS of this kind? Is it applied as signal shift or some different way? For now it seems like it is simply dropped from the consideration and only t1 is counted (TD distance).

Comment 5,6,7:

Thank you for the response and proper modification in the manuscript.
